# Relationship between the Behavior of Hydrogen and Hydrogen Bubble Nucleation in Vanadium

**DOI:** 10.3390/ma13020322

**Published:** 2020-01-10

**Authors:** Zhengxiong Su, Sheng Wang, Chenyang Lu, Qing Peng

**Affiliations:** 1Department of Nuclear Science and Technology, Xi’an Jiaotong University, No. 28, Xianning West Road, Xi’an 710049, Shaanxi, China; suzhengxiong@stu.xjtu.edu.cn (Z.S.); chenylu@xjtu.edu.cn (C.L.); 2Physics Department, King Fahd University of Petroleum & Minerals, Dhahran 31261, Saudi Arabia; qpeng@umich.edu

**Keywords:** hydrogen bubble nucleation, first-principles calculations, self-clustering, vacancy cluster

## Abstract

Hydrogen plays a significant role in the microstructure evolution and macroscopic deformation of materials, causing swelling and surface blistering to reduce service life. In the present work, the atomistic mechanisms of hydrogen bubble nucleation in vanadium were studied by first-principles calculations. The interstitial hydrogen atoms cannot form significant bound states with other hydrogen atoms in bulk vanadium, which explains the absence of hydrogen self-clustering from the experiments. To find the possible origin of hydrogen bubble in vanadium, we explored the minimum sizes of a vacancy cluster in vanadium for the formation of hydrogen molecule. We show that a freestanding hydrogen molecule can form and remain relatively stable in the center of a 54-hydrogen atom saturated 27-vacancy cluster.

## 1. Introduction

The development of fusion reactors requires a comprehensive understanding of the behavior of the structural material under irradiation. Vanadium (V) is one of the promising candidates for first wall structure support material in future fusion reactors given their outstanding mechanical performance at high temperature, superior resistance to neutron irradiation, and low activation property [1,2,3,4,5,6,7,8]. Under intense radiation by the fusion reaction, a large number of high energy protons (H^+^) bombard the first wall material, forming agglomerations and bubbles in the structural material, resulting in hydrogen-induced cracking, surface blistering and swelling [9,10,11]. These damages can seriously lead to the degradation of the mechanical properties of the material and affect the safety of the reactor operation [12,13,14]. It is generally believed that the nature of such damage phenomenon is derived from the interplay between hydrogen and various lattice defect. Therefore, it is necessary to study the behavior of hydrogen and hydrogen bubble nucleation further to slow down degradation and extend the service life of materials, particularly within the context of radiation damaged materials, such as metallic vanadium.

There are some analysis methods to study the formation of hydrogen bubbles, such as the positron annihilation technique and thermal desorption spectrometry [15,16,17], but both can only obtain partial information, and it is difficult to explain the role of hydrogen-induced bubbles. Further, hydrogen bubbles or blistering can also be observed using advanced microscopy analysis [11,18]. Still, the direct experimental observation of the initiation and the growth process of hydrogen is not readily accessible because it is related to highly localized atomic scale phenomena [19]. First-principles calculations based on the density functional theory (DFT) is probably the most appropriate method to evaluate these atomic interactions [19,20], with the result of enhancing the understanding of the involved fundamental atomic phenomena [21,22,23].

In the last decade, some theoretical analysis has found that defects such as monovacancy [23], self-interstitial atoms [24,25] and other impurity atoms [26,27,28] could act as the nucleation sites for hydrogen owing to their have the ability to block hydrogen diffusion [29,30] or trap hydrogen [31,32]. However, most of the previous first-principles calculations work is confined to the uncomplicated cases of monovacancies or divacancies, leading to the absence of research on the initial nucleation stage of hydrogen bubble. The hydrogen molecule is extremely difficult to survive in a small vacancy complex because of the strong repulsion between the hydrogen atoms, causing them to prefer to occupy interstitial sites [33]. For instance, Zhang et al. [34] explore the possibility of forming a hydrogen bubble through forcing one hydrogen molecule into the saturated monovacancy in vanadium, and suggested that the formation of hydrogen molecules would occur under sufficiently high internal pressure. Gui et al. [32] calculated the vacancy trapping behaviors of hydrogen in vanadium by two way of insertion hydrogen, the result indicated the single vacancy could accumulate as many as 12 H via simultaneous trapping. Liu et al. [22] and Hayward et al. [31] point out that the triggering conditions for hydrogen bubbles formation depend on whether the number of hydrogen atoms in the vacancy reaches a critical value. The absence of reliable physical models made it difficult to analyze the behavior of hydrogen in vacancy clusters, causing the atomic level relationship between the hydrogen bubble nucleation and hydrogen has not been completely understood and is still controversial [19].

In this study, we have deeply investigated the behavior of hydrogen in vanadium by first principles, to have further insight into the hydrogen bubble nucleation in vanadium. More precisely, we have discussed the interaction of H–H and vacancy cluster-H in detail and have connected these interactions with hydrogen bubble nucleation and blistering in vanadium. Our calculations provide the nucleation properties of hydrogen bubbles and a useful reference for vanadium as a promising candidate for first wall structural support material in future fusion reactors.

## 2. Computational Details

We perform DFT calculations using Vienna Ab initio Simulation Package [35,36]. The projector augmented wave (PAW) method was utilized to describe the electron-ion interaction [37]. The generalized gradient approximation (GGA) in the Perdew–Burke–Ernzerhof (PBE) form was applied to compute the exchange-correlation functional between electrons [38]. A (4 × 4 × 4) supercell of bcc V was employed to describe the properties of the point defect and nine-vacancy cluster, and a (5 × 5 × 5) supercell for the 27-vacancy cluster. The plane wave cutoff energy of all systems was set to 400 eV. The algorithm for structure optimization is conjugate gradient (CG), and the electronic threshold for the self-consistency cycles (SCF) was set at 10^−6^ eV. These equilibrium structures were utterly relaxed until the force acting on each atom is less than 0.001 eV/Å. The Brillouin zones were sampled with 3 × 3 × 3 k-points by Monkhorst-Pack scheme for (4 × 4 × 4) supercell [39]. As a reference, the calculated equilibrium lattice parameter of bcc V is 3.00 Å, which is consistent with previous calculations [32,33] and experimental values of 3.03 Å [40].

The binding energy between two point defects (*A*, *B*) is defined as follows [41]:(1)$$E_{b}(A,B)=E(A+B)+E(tot)-[E(A)+E(B)]$$
where E(A+B) is the energy of the structure contains *A* defect and *B* defect, E(tot) is the energy of the structure without *A* defect and *B* defect. E(A) and E(B) are the energies of the structure with *A* defect and with *B* defect, respectively. Therefore, the calculated value of negative binding energy suggests an attraction between the two defects, while a positive binding energy implies a repulsion.

## 3. Result and Discussion

### 3.1. H–H Interaction

The position of hydrogen in vanadium plays a key role because it can affect its solubility and migration. Hydrogen atoms usually sit on interstitial sites in metals. There are two symmetrical types of interstitial sites in the bcc structure: tetrahedral (T) sites and octahedral (O) sites [42]. The solution energy of hydrogen at the T sites (−0.38 eV) is lower than that of hydrogen at the O sites (−0.23 eV) by DFT calculation. Therefore, hydrogen atom prefers to occupy the T sites, in agreement with experimental observations and other DFT calculations [33,34].

Next, in order to analyze the H–H interaction comprehensively, two hydrogen atoms are placed at tetrahedral interstitial sites (T) and separated by some different distances and relax the entire system. According to Equation (1), the binding energies between the two hydrogen atoms were calculated, and the results are plotted in Figure 1.

As shown in Figure 1a, in the case that two hydrogen atoms were placed initially at 1 and 2 Å, the final relaxed distance significantly increased due to the volatile initial configuration. Therefore, the equilibrium distances for two hydrogen atoms are longer than the bond length of hydrogen molecule of 0.74 Å, which means that two hydrogen atoms cannot be directly bonded to each other to form a hydrogen molecule in bcc V. The H–H binding energies are detailed in Figure 1b, and the H–H interaction in vanadium mainly includes two parts: one is the elastic interaction due to the lattice distortion by the interstitial hydrogen, and the other is electronic interaction arising from the change of the electronic structure states of the hydrogen [43]. Both electronic and elastic interaction decrease with increasing the H–H distance. However, the electronic interaction is stronger than elastic interaction at a short distance but decreases faster with the increase of distance. The electronic repulsion plays a dominant role at short distances, which explains the phenomenon that two interstitial hydrogen atoms exhibit strong repulsion when the distances are less than 2 Å. As the distance of H–H increase from 2 to 3 Å, the elastic attraction begins to exceed the electronic repulsion. Therefore, H–H pairs show a slight attraction in this distance range. After the H–H distance increases above 3 Å, the effects of elastic and electronic interaction have reached dynamic balance, and the overall H–H interaction can be considered negligible.

In order to better understand H–H interaction, the hydrogen atoms of the projected density of states (PDOS) are calculated and presented in Figure 2. It is evident that there are two asymmetric peaks when the separation distance is small. With the further increase of separation distance, these peaks tend to merge and eventually evolve into a single peak. As the two peaks approach each other, the intensity of interaction was gradually weakened. Furthermore, the calculation indicated that two hydrogen atoms in the T sites at a separation distance of about 2.12 Å, along the [110] direction exhibited the most bound state. However, the binding energy between the two hydrogen atoms was too small (−0.035 eV) to directly bind the hydrogen atoms for a significant time.

Our present result clearly shows that hydrogen self-clustering is unlikely to occur in vanadium, which is in agreement with Takagi’s experimental work [44]. Thus, hydrogen self-clustering cannot act as seeds for hydrogen bubble growth.

### 3.2. Vacancy-H Interaction

Vacancy can act as stronger traps for hydrogen than T sites, which can accommodate several hydrogen atoms and further leads to the formation of hydrogen bubbles. To investigate the interaction between hydrogen and monovacancy in vanadium, we studied the lowest energy configurations of multiple hydrogen atoms in the monovacancy (see Figure 3). One hydrogen atom occupies a location that slightly offset by the O sites and the distance of 1.27 Å from the center of the monovacancy. The second hydrogen atom is located opposite the first hydrogen atom, which means hydrogen molecule cannot form in monovacancy. Further, the additional hydrogen atoms would preferentially occupy the O sites forming a triangle, tetrahedron, square pyramid, and square bipyramid for 3, 4, 5 and 6 hydrogen atoms, respectively [34]. In order to characterize these states quantitatively, the binding energy for the nth hydrogen atom by the most stable vac+(n−1)H structure can be defined as [45]:(2)$$E_{H}^{bind}(n)=[E(vac+nH)-E(vac+(n-1)H]-[E(V+H(T-site))-E(V)]$$
where the E(vac+nH) and E(vac+(n−1)H) are the energies of the structure with *nH* and (*n* − 1)*H* at monovacancy, respectively. The E(V+H(T−site)) is the energy of the structure with a hydrogen atom at the *T* sites. A negative value of $E_{H}^{bind}$ means that is energetically favorable to add the *n*th *H* to the vacancy with (*n* − 1)*H*. The results of our calculation are given in Table 1. Our calculation results indicate that a monovacancy can trap six hydrogen atoms at most and is consistent with other work [32].

As can be seen in Figure 3, several hydrogen atoms occupy symmetric sites and display the same PDOS curve, which means they have equivalent electronic interaction with the neighboring atoms. Then, the average binding energy is a more appropriate variable to express the environment perceive by any individual hydrogen atom [46]. Therefore, the average binding energy of all hydrogen atoms in vacancy was defined as:(3)$$E_{aver}=\frac{1}{n}[E(vac+nH)-E(vac)]-[E(V+H(T-site))-E(V)]$$

In the ion-beam analysis experiment, the binding energy between deuterium atoms and vacancy is −0.27eV [47], which is close to our calculated average binding energy of −0.29eV. Although not all hydrogen atoms are equivalent, there are at most two types of hydrogen atoms in any H-vacancy complex (see Figure 3). With the addition of hydrogen atoms in the cluster, the S-band of hydrogen atoms shifts to lower energy, along with a decrease in height and splits into two peaks or more. These trends indicate that the binding energy between hydrogen and the system is decreasing.

The hydrogen molecules within the bubble are essential for the structural properties of vanadium. However, there is a lack of research about the hydrogen trapping in the vacancy cluster and how the hydrogen molecules appear in the vacancy cluster. To further investigate the interaction of H-vacancy and the details of hydrogen bubble growth, we explored the minimum sizes of a vacancy cluster in vanadium for the formation of hydrogen molecule. We investigate the several sizes of vacancy cluster, ranging from a single vacancy, to nine vacancies, to twenty-seven vacancies of a (2 × 2 × 2) cube whose vertex is truncated. The reason is that they close to be spherical and tend to maintain the lowest energy even after adding hydrogen atoms [46]. After saturating the vacancy cluster with hydrogen atoms [22], we examine whether the hydrogen molecule can be stably located in the center of the vacancy cluster.

Our calculations have shown that a monovacancy in vanadium can trap six hydrogen atoms at most on the inner surface. If we force a hydrogen molecule into the vacancy coated by six hydrogen atoms, the average binding energy of all hydrogen is 0.50 eV, which indicates that the relaxed configuration is much energetically unfavorable. And the bond length of the hydrogen molecule would become 0.791 Å after relaxation, 7% larger than the bond length of the hydrogen molecule (0.75 Å) in a vacuum. Therefore, hydrogen molecule cannot form in monovacancy.

There are six facets with four hollow sites on each facet and eight corners in a nine-vacancy cluster. There is no doubt that it would be a time consuming and tedious work when hydrogen atoms are placed one after another into this vacancy cluster and minimize the energy to find the optimal structure of the entire system at each step. We assumed that the location of the hydrogen atom is equivalent in all facets due to the high symmetry of the vacancy cluster shape [48]. Further, we noticed that these facets are made up of a small chip of {100} planes in bcc V, and hydrogen atoms were preferentially located in the hollow sites on V (100) at a monolayer coverage [49]. Therefore, we fill all four hollow sites with hydrogen atoms, and there are no suitable interstitial sites in the facets for the further addition of hydrogen atoms due to the repulsion with each other. The structure of the nine-vacancy cluster occupied by twenty-four hydrogen atoms on the inner surface is shown in Figure 4a. We introduce a hydrogen molecule into the center of the 24H-saturated vacancy cluster and optimize the structure of the entire system. Then the average binding energy for all hydrogen atoms in the structure is calculated as −0.22 eV. Moreover, it was found that the H–H length of the hydrogen molecule is 0.78 Å, slightly larger than 0.75 Å of a hydrogen molecule in a vacuum [50]. The result means that the freestanding hydrogen molecules are absent in a nine-vacancy cluster.

The next step is to examine a 27-vacancy cluster by following the same method used for the 9-vacancy cluster. However, there are still some differences in their structure. Not only did the 27-vacancy cluster have six {100} facets with five hollow sites on each facet, but also the eight {111} facets have three bridge sites on each facet. Accordingly, we also need to consider the distribution of hydrogen atoms on the {111} facet. It was found that hydrogen is most willing to occupy the bridge sites on the {111} facet by first principles calculation. In consideration of the above two cases, there are 54 hydrogen atoms occupied on the inner surface of 27-vacancy cluster in total, including thirty hydrogen atoms in {100} facet and twenty-four hydrogen atoms in {111} facet. The structure of the 27-vacancy cluster occupied by fifty-four hydrogen atoms on the inner surface is shown in Figure 4b. Similarly, a hydrogen molecule was introduced to the center of the 54H-saturated vacancy cluster, and the structure of the entire system was further optimized. The average binding energy for all hydrogen atoms is −0.23eV, and the H–H bond length of this hydrogen molecule is the same as that of the hydrogen molecule in a vacuum. Moreover, the system energy was slightly changed when the orientation of hydrogen molecule from [100] to [111], which means the hydrogen molecule is relatively stable in the vacancy cluster center.

Furthermore, the hydrogen atoms on the inner surfaces of vacancy and the hydrogen in molecular are considered to be significantly different in terms of PDOS (Figure 4). The former shows a strong hybridization with the neighboring vanadium. In the nine-vacancy cluster structure, the PDOS of the molecular hydrogen has two peaks, and the apparent overlap at low energy peak is due to the interaction with neighboring vanadium atoms and hydrogen atoms on the inner surface. In the 27-vacancy cluster structure, there is only a single distinct peak indicating that the orbital of hydrogen molecule is barely damaged. These results further confirm that a freestanding hydrogen molecule could form and be relatively stable in the center of this cluster. The above analysis suggests that hydrogen molecule could form only when the vacancy cluster is large enough and saturated with hydrogen.

## 4. Conclusions

In summary, we have deeply investigated the electronic structures of defect bonding with hydrogen in vanadium using first-principles calculation. There is only a fragile attraction between the hydrogen with other interstitial hydrogen atoms, implying that it cannot directly bind the hydrogen atoms for a significant time. As a result, hydrogen self-clustering is unlikely to occur in bulk vanadium, consistent with previous experimental studies. We focus on whether or not the hydrogen molecule would have the opportunity to form in these configurations, rather than how such a configuration would come to exist. Therefore, this spherical 27-vacancy cluster is considered to be the smallest place for hydrogen bubble formation in vanadium.

In conclusion, our model provides a framework for characterizing a vacancy cluster as hydrogen trapping. The information obtained from this study and the proposed model can be used as input parameters for further atomic simulations, such as LAMMPS or KMC models, and help us further study the thermodynamic properties of hydrogen bubbles in a small void. There are some limitations of the model and assumptions which may affect the exact amount of hydrogen atoms that can saturate in the vacancy cluster. However, in real physical systems, it is generally not necessary to know exactly how many hydrogen atoms in vacancy cluster saturations are needed. The number of hydrogen atoms is sensitive to temperature and environmental alterations in the microstructure. Our results and model could serve as a general method to rapidly determine whether hydrogen molecules are likely to form in the vacancy cluster. We described the conditions under which we expect to characterize the existence of hydrogen molecules in experiments, providing support for the observation results of the experiment. The present theoretical result has pushed forward the understanding of hydrogen mechanisms affecting the nucleation of bubble and microstructure evolution in vanadium at the atomic level.

## Figures and Tables

**Figure 1 materials-13-00322-f001:**
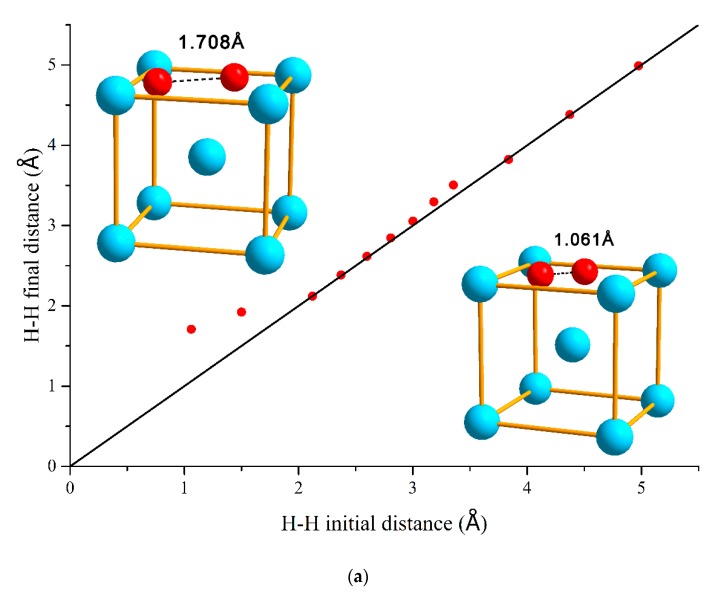
(**a**) Initial and final distances between H–H pairs of atoms. The unit cell in the lower right and upper left represents the schematic of the H–H initial distance of 1.061 Å and the final distance of 1.708 Å after optimization, respectively. (**b**) The interaction energies of H–H pairs as a function of the H–H distance in bcc V. The unit cell represents a schematic of the strongest bonding energy in H–H configuration.

**Figure 2 materials-13-00322-f002:**
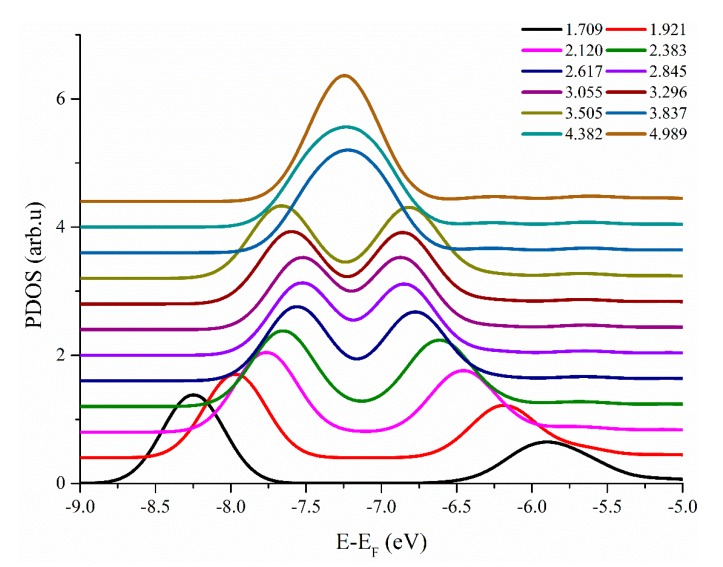
The projected density of states (PDOS) of the hydrogen atoms in different distances of H–H pairs in vanadium. The numbers in the legend represent the distance of H–H pairs after relaxation. In order to see more clearly the changes in each curve, each line is separated from top to bottom.

**Figure 3 materials-13-00322-f003:**
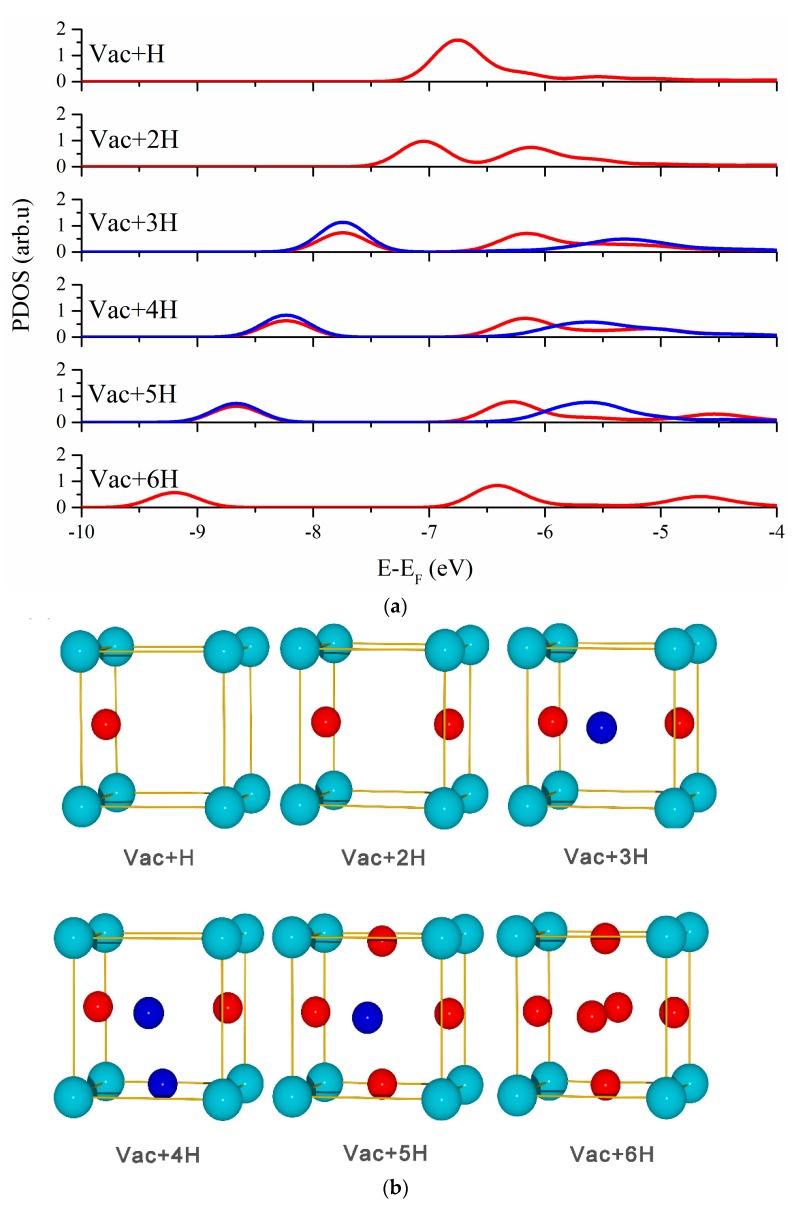
(**a**) The PDOS curves and structures of multiple hydrogen atoms in the monovacancy in bcc vanadium. (**b**) The two types of hydrogen atoms are in blue and in red, respectively, corresponding to the color of the PDOS curves in the above.

**Figure 4 materials-13-00322-f004:**
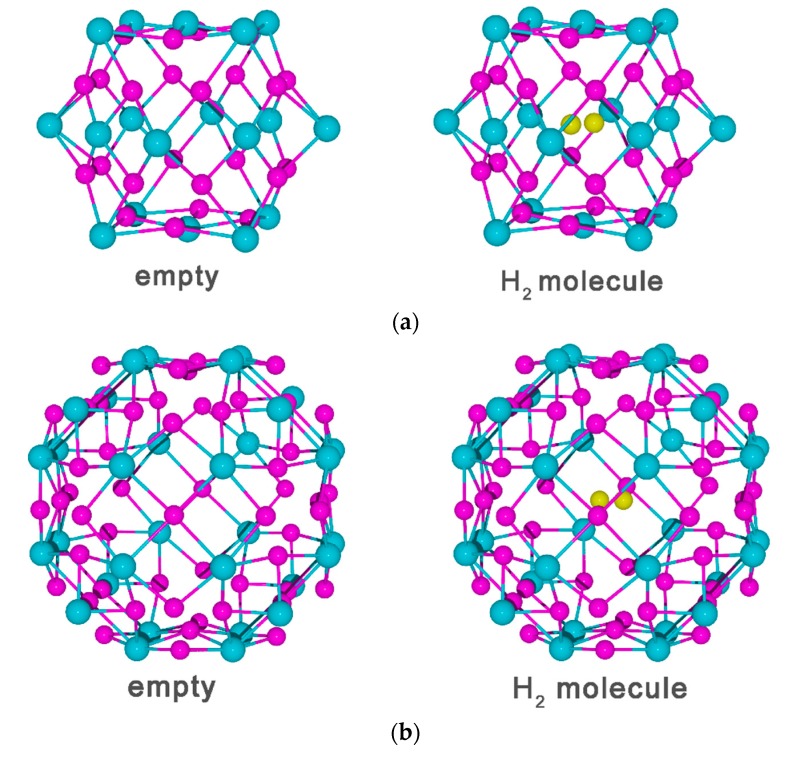
There are two types of schematic H-saturated vacancy cluster structure: (**a**) The 9-vacancy cluster of 24H-saturated in a (4 × 4 × 4) supercell vanadium (**b**) The 27-vacancy cluster of 54H-saturated in (5 × 5 × 5) supercell vanadium. Vanadium atoms are in blue, saturated H in pink and molecular H in yellow. Only the V atoms on the surface of the vacancy cluster are displayed. The PDOS for hydrogen on the inner surface and in the molecule, and neighboring vanadium corresponding to the two types of structures with hydrogen molecule are shown on (**c**,**d**), respectively.

**Table 1 materials-13-00322-t001:** The binding energy for the nth hydrogen atom by the most stable vac+(n−1)H structure in bcc vanadium.

Configuration	$E_{H}^{bind}\;(\mathrm{eV})$	Ref. [34] (eV)
vac+H	−0.34	−0.31
vac+2H	−0.43	−0.40
vac+3H	−0.27	−0.25
vac+4H	−0.28	−0.25
vac+5H	−0.24	−0.24
vac+6H	−0.16	−0.12
vac+7H	0.65	0.69

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
