# Peer review of "Relationship between the Behavior of Hydrogen and Hydrogen Bubble Nucleation in Vanadium"

_materials, 2020, doi:10.3390/ma13020322_

Round 1

Reviewer 1 Report

The authors of the manuscript are providing the computational modeling on hydrogen impurities in metallic vanadium and consider interstitial H and cluster of hydrogens in monovacancy/cavity of vacancies in vanadium. 

The topic of the study is of immediate interest for the publication in the Materials journal. However the quality of the manuscript is not enough for publication. The manuscript requires significant improvement in many ways.

The Introduction of the manuscript consists of only one paragraph and not explores many aspects of the topic. In particular, the authors should address the question of the source of hydrogens. Is it the molecular hydrogen or it comes from the high-energy protons? In experiment by Rohrig 1998 (ref.9 of the manuscript) the source of hydrogen is molecular hydrogen. In the manuscript the authors consider the atomic hydrogens and place them in the appropriate places into metallic vanadium. It is necessary to provide the physical validation of such choice.
In the paper by Troev et.al. (Nuclear Instruments and Methods in Physics Research B 248 (2006) 297) it is considering similar theoretical model, but the authors of the manuscript did not include this paper into Introduction.

It is necessary to validate the theoretical model for calculations. For the considered model of 4x4x4 supercell of 2 atom primitive cell of bcc vanadium, the supercell will have totally 128 atoms. Hydrogen impurities in interstitial positions or adsorbed in cavity will have some dipole configuration. The interaction of these periodic dipoles in a such small system will be very strong and completely destroy the real physical picture. Troev et.al. (2006) provide the estimates for inaccuracy of calculation in a such system, and concluded that for a such system the supercell of 2000 will be enough to produce inaccuracy of calculation of 0.01 eV.

On lines 156 and 172 the authors concluded that the hydrogen molecule is not stable based on the fact that the bond length of the H2 species is 0.79 A instead of 0.75 A as for free H2 molecule. This is wrong conclusion. This means only that H2 species adsorbed to the vanadium, which slightly elongate the H-H bond. See for example https://doi.org/10.1103/PhysRevB.72.155404     

Reviewer 2 Report

The density functional theory provides a universal and reliable tool for prediction of a variety of properties of solids. One of the applications is the calculation of defect formation energy for single defects and the binding energy for the clusters of defects. Such theoretical results can be of great importance form the viewpoint of modern materials science. The present paper presents a computational study of hydrogen defects in bcc vanadium, which constitutes important material for fusion reactors construction. The creation of various hydrogen defects may lead to deterioration of the material. The study is based on a reliable software and clear methodology and is aimed at discussing the possible nucleation of a hydrogen bubble form the atomic point of view.

The Authors have calculated first the interaction energy (binding energy) for point H defects as a function of their distance. Then the issue of stability of H clusters formed in a monovacancy in bcc vanadium is discussed. The binding energies of last hydrogen atom forming a cluster of given size in monovacancy are calculated. Also, the stability of a hydrogen molecule inside a vacancy with numerous H atoms saturated at the edges is investigated.

The paper is in general sound and well written, the discussion is rather clear and convincing. The results are consistent and of interest to the Readers. I recommend the manuscript for publication in Materials, provided that the Authors give prior consideration to a few minor points listed below (mainly related to the data presentation):

• In the part ‘Computational details’ – what was the algorithm used for relaxation of the atomic positions (structure optimization)? What was the energy threshold for self-consistency used in the calculations?

• Fig. 2 and Fig. 4: please make label to horizontal axis more readable: is it E-E_F?

• Fig. 3: please provide labels for horizontal and vertical axis.

• If units for PDOS in Fig. 2,3,4 are arbitrary units, please use “arb. u.”

• Regarding the positions of the interstitial H point defects mentioned in page 2, line 67/68 – are the H atoms considered in tetrahedral positions (denoted by T)? Maybe it would be instructive to draw schematically the positions in the unit cell of bcc vanadium, indicating a few shortest distances between the H atoms in tetrahedral positions. 

• Does the relaxation of the entire system mentioned in page 2, line 68 mean the relaxation of the atomic positions (without change of the lattice constants)? This could be described explicitly.

• In page 2, line 69, the sentence is imprecise – the binding energies are calculated according to Eq. (1), but the distances are not.

• For close H neighbours (initial distance < 2 A) the initial and final distance is clearly very different, as it is evident form Fig. 1(b). Do both initial and final positions correspond to tetrahedral sites or not? Could the Authors draw the final positions of both H atoms in the unit cell?

• In Fig. 1(b), drawing the horizontal line at zero energy might be useful.  In the caption of Fig.1-in part (b) is the distance the final distance between H atoms?

• Page 2, equation (1): it might be useful from the point of view of the reader to cite here some reviews related to calculations of the formation energy and binding energy of point defects and clusters of defects – for example DOI: 10.1063/1.1682673 and DOI: 10.1103/RevModPhys.86.253. Actually, the binding energy is defined using the eq. (14) in the paper DOI: 10.1063/1.1682673 on the basis of the formation energies (which, in turn, are defined using eq. (2) in the same work – but finally the formula for binding energy reduces to the expression used by the Authors).

• Some language corrections would be advised, the examples are:

Page 6, line 195: probably at the end of figure caption” “are shown, respectively”.

Page 6, line 203: “be relatively stable”.

Page 7, line 201: “experimental studies”.

Reviewer 3 Report

The authors theoretically investigated the nucleation of hydrogen impurities in bulk vanadium. Employing the ab-initio simulations based on the density functional theory, the authors evaluated the stability of the insertion of hydrogen impurities in vanadium with/without vacancy. The methods used in the present work technically sound. However, the results are not significant enough to develop a further understanding of the nucleation of hydrogen impurities in bulk vanadium. Thus, I could not recommend this manuscript for the publication from Materials. I describe a couple of comments to back up my conclusion below.

1) Significance

The authors investigated the stability of the insertion of hydrogen impurities in bulk vanadium for different configurations. However, the novel understanding of the nucleation of hydrogen impurities has not been developed with their results, although it hast to be a key point of the present work. Thus, I recommend the authors to clearly highlight the novel knowledge developed by the present work in the manuscript.

2) Figures

In the present manuscript, the quality of figures is not enough for publication. In some figures, labels are missing or too small to read. In some figures, units of the axis are missing.

3) English

In the present manuscript, the quality of English is not enough for publication because it is difficult to understand some parts of the manuscript due to wrong grammar or typos. So, I would recommend the authors to improve their manuscript such that readers can properly understand the results and the conclusion of the manuscript.

Round 2

Reviewer 1 Report

The authors made significant improvement of the manuscript and took into account all suggestions and recommendations of the reviewers. The manuscript can be published in the present form after the minor corrections:

lines 25-26: change "protons/hydrogen (H) atoms" -> "protons (H+)" line 26: change "platelets" -> "agglomerations" or "clusters" line 50: completely rewrite the part of the sentence: "For instance, Zhang et al. [34] through force on one hydrogen molecule into the saturated monovacancy in vanadium to explore the possibility of forming hydrogen bubble, ..." because the meaning of this phrase is missed, probably it is necessary to insert some verb here " Zhang et al. [34] ... through" line 120: to correct "E_F" to make F as subscript and remove underwriting line 203: to insert the verb into the sentence: "The result means that the absence of freestanding hydrogen molecule in a 9-vacancy cluster." line 254: remove one word "however" or "indeed" in the phrase: "However, indeed ..."

Author Response

Response to Reviewer 1 Comments

Thank the reviewers for these precious comments concerning my manuscript entitled “Relationship between the behavior of hydrogen and hydrogen bubble nucleation in vanadium” (ID: materials-673919 ). These comments are valuable and very helpful for revising and improving our paper.

We have studied comments carefully and have made corrections which we hope meet with approval. A point by point response to the reviewer’s comments are as following:

Point 1: lines 25-26: change "protons/hydrogen (H) atoms" -> "protons (H+)"; line 26: change "platelets" -> "agglomerations" or "clusters".

Response 1: We have revised the sentence according to the Reviewer's suggestion.

Point 2: line 50: completely rewrite the part of the sentence: "For instance, Zhang et al. [34] through force on one hydrogen molecule into the saturated monovacancy in vanadium to explore the possibility of forming hydrogen bubble, ..." because the meaning of this phrase is missed, probably it is necessary to insert some verb here " Zhang et al. [34] ... through"

Response 2: According to the Reviewer's suggestion, the sentence has been modified to “ For instance, Zhang et al. [34] explore the possibility of forming hydrogen bubble through force on one hydrogen molecule into the saturated monovacancy in vanadium,…”

Point 3: line 120: to correct "E_F" to make F as subscript and remove underwriting.

Response 3: We have revised all the relevant symbols according to Reviewer’s suggestion.

Point 4: line 203: to insert the verb into the sentence: "The result means that the absence of freestanding hydrogen molecule in a 9-vacancy cluster."

Response 4: According to the Reviewer's suggestion, the sentence has been modified to “The result means that the freestanding hydrogen molecules are absent in a 9-vacancy cluster.”

Point 5: line 254: remove one word "however" or "indeed" in the phrase: "However, indeed ..."

Response 5 : We have remove the word “indeed” according to Reviewer’s suggestion.